# A systematic machine learning and data type comparison yields metagenomic predictors of infant age, sex, breastfeeding, antibiotic usage, country of origin, and delivery type

**Alan Le Goallec**[1,2‡], **Braden T. Tierney**[1,3,4,5‡], **Jacob M. Luber**[1], **Evan M. Cofer**[6], **Aleksandar D. Kostic**[3,4,5]*, **Chirag J. Patel**[1]*

1 Department of Biomedical Informatics, Harvard Medical School, Boston, Massachusetts, United States of America, 2 Department of Systems Biology, Harvard University, Cambridge, Massachusetts, United States of America, 3 Section on Pathophysiology and Molecular Pharmacology, Joslin Diabetes Center, Boston, Massachusetts, United States of America, 4 Section on Islet Cell and Regenerative Biology, Joslin Diabetes Center, Boston, Massachusetts, United States of America, 5 Department of Microbiology and Immunobiology, Harvard Medical School, Boston, Massachusetts, United States of America, 6 Lewis-Sigler Institute for Integrative Genomics, Princeton University, Princeton, New Jersey, United States of America

‡ These authors share first authorship on this work.
* aleksandar.kostic@joslin.harvard.edu (ADK); chirag_patel@hms.harvard.edu (CJP)

**Data Availability Statement:** All raw sequencing data are available from the Diabimmune project site (https://pubs.broadinstitute.org/diabimmune) and

## Abstract

The microbiome is a new frontier for building predictors of human phenotypes. However, machine learning in the microbiome is fraught with issues of reproducibility, driven in large part by the wide range of analytic models and metagenomic data types available. We aimed to build robust metagenomic predictors of host phenotype by comparing prediction performances and biological interpretation across 8 machine learning methods and 4 different types of metagenomic data. Using 1,570 samples from 300 infants, we fit 7,865 models for 6 host phenotypes. We demonstrate the dependence of accuracy on algorithm choice and feature definition in microbiome data and propose a framework for building microbiome-derived indicators of host phenotype. We additionally identify biological features predictive of age, sex, breastfeeding status, historical antibiotic usage, country of origin, and delivery type. Our complete results can be viewed at http://apps.chiragjpgroup.org/ubiome_predictions/.

## Author summary

The human microbiome is hypothesized to influence human phenotype. However, many published host-microbe associations may not be reproducible. A number of reasons could be behind irreproducible results, including a wide array of methods for measuring the microbiome through genetic sequence, annotation pipelines, and analytical models/prediction approaches. Therefore, there is a need to compare different modeling strategies and microbiome data types (i.e. species abundance versus metabolic pathway abundance) to determine how to build robust and reproducible host-microbiome predictions. In this

EBI's sequence archive (accession = ERP005989; https://www.ebi.ac.uk/metagenomics/studies/ERP005989).

**Funding:** Microsoft Azure, Harvard Research Computing, Amazon Web Services provided compute resources for this work to CJP. This research was additionally supported by the National Institutes of Health (nih.gov): T32 DK110919 to JL, R00ES23504 to CJP, R21ES205052 to CJP, R01AI127250 to CJP), the National Science Foundation (nsf.gov): 1636870 to CJP, the American Diabetes Association (ADA) Pathway to Stop Diabetes Initiator Award (diabetes.org): #1-17-INI-13 to AK, Smith Family Foundation Award for Excellence in Biomedical Research (https://www.smithfamilyfoundation.net/health/) to AK. E.M.C. was supported by National Institutes of Health (NIH) grant T32 HG003284 and the National Science Foundation Graduate Research Fellowship Program (NSF-GRFP). The funders played no role in study design, data collection and analysis, decision to publish, or preparation of the manuscript.

**Competing interests:** I have read the journal's policy and the authors of this manuscript have the following competing interests: CJP is a founder and advisor at XY.ai. ADK is a founder of and advisor at DeepBiome Therapeutics.

work, we executed a broad comparison of different predictive methods as a function of microbiome data types to effectively predict host characteristics. Our pipeline was able uncover robust microbial associations with phenotype. We additionally recommended considerations for reproducible microbiome-host association pipeline development. We claim our work is a necessary stepping stone in increasing the utility of emerging cohort data and enabling the next generation of efficient microbiome association studies in human health.

## Introduction

With advancements in sequencing and machine learning, the number of available microbiome analytic tools and data types (e.g. species/genus abundance, metabolic pathway abundance) for microbiome analysis has proliferated. On the one hand, these developments hold immense promise–variations in the human microbiome are associated with host health and environment, and the more ways we can understand the microbiome, the better we will be able to leverage its use in the clinic [1,2]. Changes in the microbiome have demonstrated classification efficacy for a range of human diseases, like type 2 diabetes and colorectal cancer [3,4]. However, unlike other aspects of microbiome research, including experimental design and sequencing processing, there are limited codified "best practices" for connecting or associating the microbiome with host phenotype, with diverse documented approaches that range from simple non-parametric statistical tests (i.e. Wilcoxon tests) to complex machine learning (i.e. random forests) [5,6,7]. Therefore, the field often faces problems of reproducibility and difficult-to-interpret results driven by increased variation in methods and study design [8]. For example, Forslund et al demonstrated in a meta-analysis of Type 2 Diabetes—microbiome associations that results were in large part dependent on whether or not a given study adjusted for metformin usage [9].

The motivation for the work described here stems from the challenge of generating reproducible predictive models for host phenotype that use, at least in part, microbiome information as input features. Generalizable and robust modeling is essential for the microbiome to achieve clinical diagnostic utility. In observational studies, it is hypothesized that "Most Research Findings are False" [10]. One source of false findings includes variation in study design, such as choice of model. Variation in findings have been described as a "Vibration of Effects," (VoE) and it has been shown to drastically affect the direction of the relationship between dependent and independent variables [11].

In this work, we quantify VoE due to algorithmic and data type choice. We tested the performance of 8 standard machine learning algorithms on a wide array of (a) host demographic and phenotypic data and (b) microbiome data types, including reference-based microbiome data and *de novo* assembled microbiome data (For a full table of definitions, please see S1 Table). Many microbiome studies, such as Microbiome-Association-Studies (MAS), utilize a range of associative tools, like linear regression or random forests [12]. Microbiome studies are also often executed using species-level or "reference-based" approaches, which are inherently limited by their using only known sequence information. *De novo* assembly-derived, gene-level data, on the other hand, extracts probable genetic elements from sequencing information without the use of any reference information. *De novo* based approaches are not limited by the known universe of microbial genomics, and can identify the "dark matter" of metagenomics, undiscovered genes [13]. However, working with genes–instead of the species that contain them–could add noise (such as measurement error), leading to both loss of power

and spurious findings. Further still, the predictive capability of the multitude of algorithmic scenarios are often reported without comparison to a "baseline", such as a simple linear regression that models a phenotype as a function of demographic characteristics (e.g. age, sex).

Biologically speaking, our analyses here additionally quantified the relative capability of demographic data alone and the microbiome to predict six demographic features (which we also refer to as phenotypes): infant age, sex, breastfeeding status, prior antibiotic usage, delivery type, and home country. These phenotypes have been studied to varying degrees in the literature, with many studies reporting associations with breastfeeding and delivery type [14–19]. Additionally, de la Cuesta-Zuluaga et al showed that alpha diversity varies with sex across age in adults, Subramanian et al and Galkin et al have built a predictors of age in children and adults, respectively, and Jernberg et al identified the presence of long term effects of antibiotic usage on the microbiome [20–23]. In additional results stemming from our model and data type comparison, we build on this existing body of literature, identifying novel associations with, specifically, age, sex, delivery type, and antibiotic usage. Determining microbial associations with these and other phenotypes is crucial for our understanding of how the microbiome changes in males and females as they age in this critical window of development in early life. Understanding how the microbiome varies in association to features like age and sex will quantitate the degree of potential confounding by demographic variables in other microbiome-association studies.

## Results

### Predictive performance varies greatly depending on algorithm and microbiome data type

Our complete results, as well as a full exploration of the relationships between our demographic variables and cohorts, can be viewed at http://apps.chiragjpgroup.org/ubiome_predictions/. This resource includes the associations between and distributions of features used in training (e.g. age and sex), top microbiome predictors of phenotypes, outcomes for specific models and datatypes, and detailed model training/output information (e.g. sensitivities, specificities, and model hyperparameters).

We aggregated and processed (Fig 1A) 1,570 human samples from four different studies, each with repeated microbiome samples from infants from four disparate European countries (Sweden, Finland, Russia, and Estonia) in the first three years of their lives [24–27]. We selected these datasets due to their 1) longitudinal nature 2) similar and high-quality metadata and 3) cohort size (>50 samples). It is worth nothing that preliminary principal component analysis did identify some stratification between datasets on the basis of metadata alone and by microbiome data type, particularly between the Swedish (Bäckhed_2015) cohort and the others (S1 Fig). Using a reference-based method, we identified 771 microbial species across all samples. With *de novo* assembly and microbial gene prediction, we identified and quantified the abundance of 9,903,745 microbial genes. In accordance with the literature, to reduce the dimensionality of our gene data, we clustered highly correlated genes into 8,302 Co-Abundance Groups (CAGs) with sizes ranging from 2 to 22,563 genes (S2 Fig) [28]. We additionally collapsed our *de novo* assembled genes into pathways via alignment to the BioCyc Tier 3 database.

For prediction, we used 2 different elastic net (Elastic Net and Elastic Net 2) implementations, 2 random forest (RF and RF2) implementations, 2 gradient boosted machine (GBM and GBM2) implementations, support vector machines (SVM, kernels: linear, polynomial of degree 2 and radial), K-nearest neighbors (KNN) and naive Bayes (NB) to predict/classify infant age, sex, country of origin, delivery type (Cesarean or vaginal), breastfeeding status, and antibiotic usage (Fig 1B). For a high-level overview of these modeling strategies, please see the S1 Text. We identified limited class imbalance in our phenotypes of interest, with the

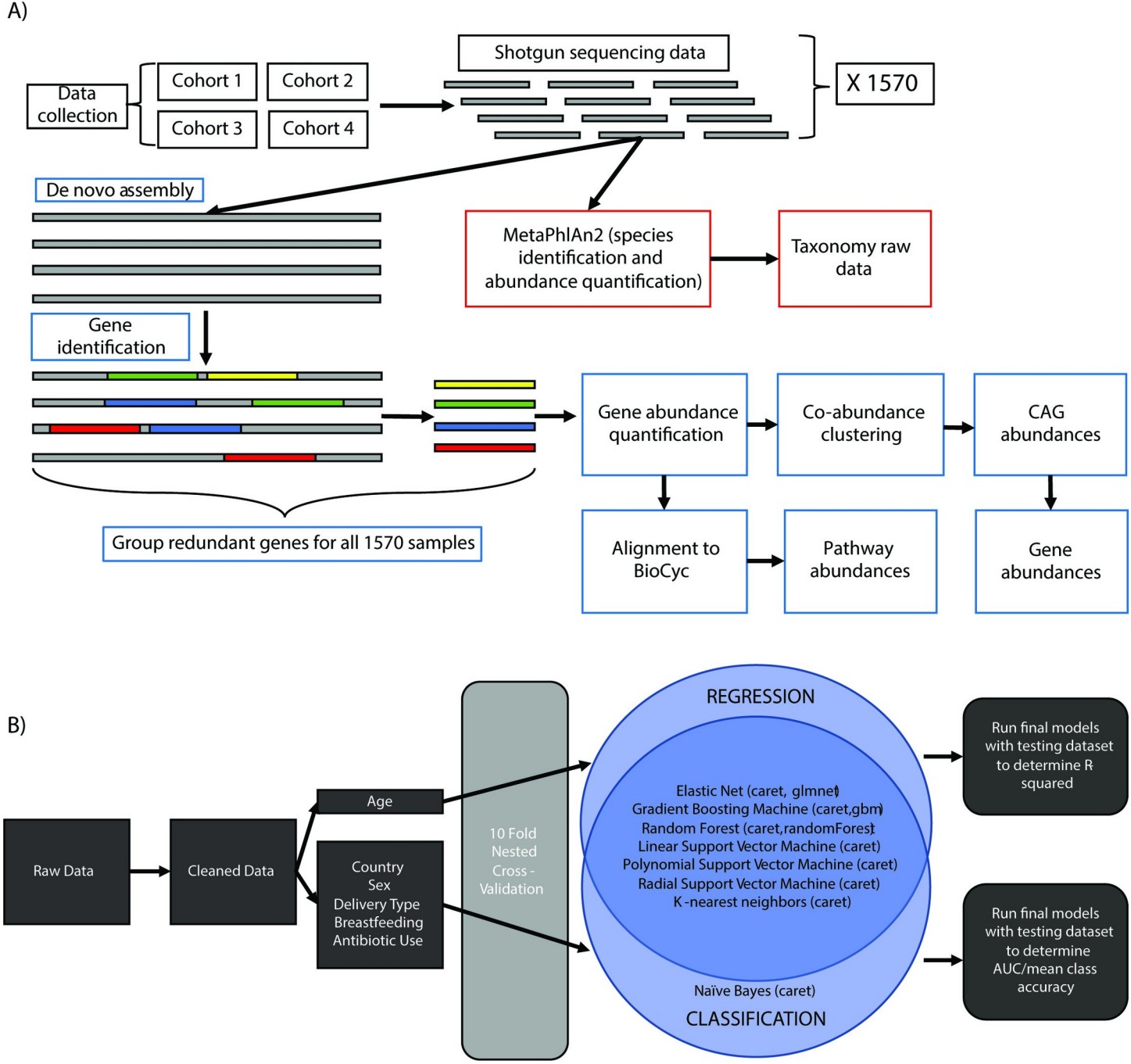

**Fig 1.** A) Data/Feature processing pipeline. We aggregate our data, and for each sample we identify the abundance of each species found within it via MetaPhlan2. We *de novo* assemble each sample and identify the non-redundant set of microbial genes within them. We quantify and normalize the abundance of each gene and then cluster them based on co-occurrence into CAGs. We then collapse raw genes into BioCyc pathways. Finally, we extracted genes for modeling from phenotype-associated-CAGs. B) Machine learning pipeline. Raw data is cleaned according to phenotypic variable completeness. We then use a nested cross-validation and a suite of machine learning tools to run our prediction analysis.

exception being delivery type (Total cesarean samples = 132, Total vaginal delivery samples = 1,185, S2 Table). For each model, we used 10x10 fold nested cross validation (S3 Fig), specifying that all samples from a given individual were present within the same fold. We tested

**Table 1. Best performing machine learning algorithms on the testing set for both experimental groups (including microbiome data) versus control group (just demographic data).**

|  | Metric | Best Predictor Set | Best Algorithm, Experimental | Best Experimental Algorithm, Performance | Best Algorithm, Baseline | Baseline Performance | Difference b/w Best Experimental and Control Performance Metrics |
|---|---|---|---|---|---|---|---|
| **Age** | R-Squared | CAGs + Demographics | Random Forest (Caret) | .625+-.021 | Random Forest 2 | .120+-.013 | 0.505 |
| **Antibiotic Usage** | AUC of the ROC | Genes + Demographics | Elastic Net (Caret) | .796+-.013 | Random Forest 2 | .786+-.013 | 0.01 |
| **Exclusively Breastfed** | AUC of the ROC | Genes + Demographics | Gradient Boosted Machine (Caret) | .794+-.012 | Gradient Boosted Machine (Caret) | .786+-.013 | 0.008 |
| **Delivery Type** | AUC of the ROC | Genes + Demographics | Elastic Net (Caret) | .760+-.021 | Gradient Boosted Machine 2 | .587+-.025 | 0.173 |
| **Sex** | AUC of the ROC | Genes | Gradient Boosted Machine (Caret) | .605+-.016 | Naive Bayes | .529+-.019 | 0.076 |
| **Country of Origin** | Mean Class Accuracies | Genes + Demographics | Gradient Boosted Machine 2 | .807+-.012 | Gradient Boosted Machine (Caret) | .651+-.014 | 0.156 |

5 data types (and combinations therein) for each outcome of interest. These were host demographics (which we refer to as our "baseline" data) and each microbiome data type: pathway relative abundance, CAG relative abundance, MetaPhlAn2 taxa relative abundance, and gene relative abundance.

After we processed metagenome information for each sample (See Methods), we input the 3 different types of the cohort to the machine learning pipeline (Fig 1A and 1B). We measured performance with a nested cross-validated $R^2$ or AUC. We estimated the error on the performance by bootstrapping (See Methods). The most common optimal algorithms were GBMs, but overall, the results of our experiments varied depending on phenotype, data type, algorithm, and algorithm hyperparameters (Table 1, S3 Table). In some cases, such as for age, linear algorithms (the elastic net) outperformed the non-linear ones (e.g. random forests). Even between the baseline (demographics only) data and the experimental (demographics/microbiome combined) data for the same target phenotype, there wasn't a consensus "best" algorithm. As observed in Table 1, there was indeed no significant difference between the baseline and microbiome-inclusive models for antibiotic usage and for exclusive breastfeeding. Other demographic variables were highly predictive: for example, the maximum AUC for demographic variables along predicting antibiotic usage was 0.786+-0.03. Most notably, country of origin and age at collection were the most predictive of our other phenotypes (though their relative importance varied depending on algorithmic choice). On contrary, significant differences were observed for age, delivery type and sex, which shows that the extra predictive accuracy comes from the microbiome variables and cannot be explained by cohort differences. Each algorithm, with the exception of the SVMs, was the best performer for at least one of the variables tested. Algorithm performance, such as for the elastic net, at times varied greatly between folds. For all phenotypes except age and sex, the mixture of genes and demographics outperformed all other data types. For sex, genes alone were most effective at distinguishing between males and females. Pathway-level annotations were never the best performing predictors for any phenotype.

## The influence of data type and algorithm choice on concordance between predictor importance

In an effort to inform future reproducible microbiome modeling, we sought to identify which data types, (between CAGs, MetaPhlan2 taxonomies, or pathways) and algorithms produced

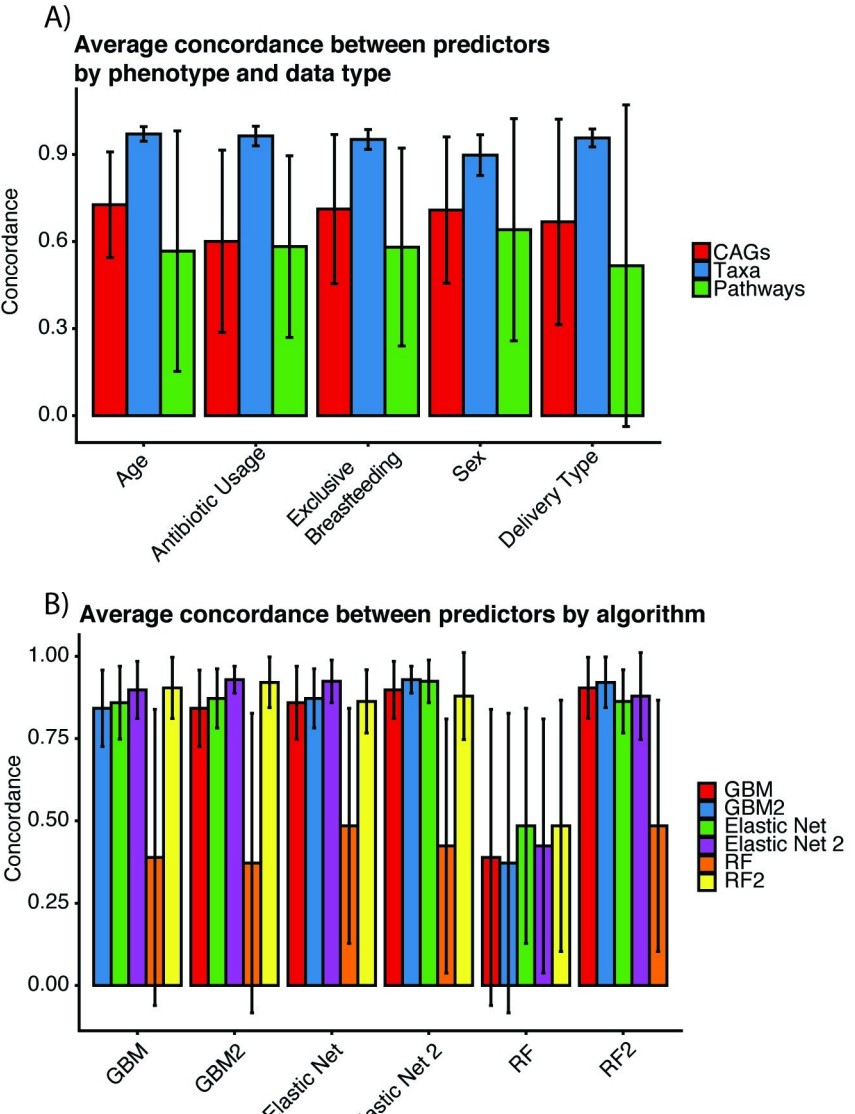

**Fig 2.** Concordance between most important predictors, measured by similarity in ranking and relative importance, between A) phenotype and data type and B) algorithm choice.

the most concordant ranking of predictors in terms of relative importance (Fig 2). That is to say, we aimed to test if algorithms agreed in terms of which microbial features were the most relevant to prediction. We measured concordance via Spearman correlation between microbiome feature rankings. We chose not to investigate genes as, due to our methods in identifying phenotype-specific gene predictors from different CAGs, each phenotype was not analyzed with the same set of genes (See Methods). We found data type to play a substantial role in the variability of predictor importance, with MetaPhlAn2 taxonomies being the most consistent (average Spearman: 0.95) across models, and CAGs being the least consistent (average Spearman = 0.68). We did not find concordance in predictor relative importance to be heavily contingent on phenotype.

Caret's EN implementation produced the ordering of predictors most consistent with all other algorithms (average Spearman rho = 0.81). Caret's Random Forest produced the least concordant results compared to all other algorithms (average Spearman rho = 0.43). The Random Forest implementation from the randomForest package was on average nearly twice as concordant with the other algorithms, demonstrating the immense impact hyperparameters and hyperparameter tuning methods have on algorithmic performance.

## Predictive performance for breastfeeding status, antibiotic usage, delivery type, sex, and country of origin

*De novo* predicted genes outperformed reference-based taxonomies for all classification tasks (Table 1, Fig 3). Although genes were the strongest performing microbiome feature, the ranking of the other data types were not consistent for each phenotype. For example, for antibiotic usage (best model: elastic net (Caret); best data type: genes + demographics; max AUC experimental = 0.796+-0.013, max AUC baseline = 0.786+-.013) and breastfeeding (best model: gradient boosted machine (Caret); best data type: genes + demographics; max experimental AUC = 0.794+-0.012, max baseline AUC = 0.794+-0.013), the models that used CAGs, genes, or pathways had on average lower AUCs than the baseline models. Notably, these two models had the lowest change between baseline and the best performing model overall, demonstrating a modest impact of the microbiome on classification accuracy.

These results stand in contrast to those for sex (best model: gradient boosted machine (Caret); best data type: genes; max experimental AUC = 0.605+-0.016, max baseline AUC = 0.529+-0.019), country of origin (best model: gradient boosted machine 2; best data type: genes + demographics; max experimental mean class accuracies = 0.807+-0.012, max baseline mean class accuracies = 0.651+-0.04), and delivery type (best model: elastic net (Caret); best data type: genes + demographics; AUC experimental = 0.760+-0.021, AUC baseline = 0.587+-0.025). These three tasks were, on average, bolstered by the addition *de novo* assembled microbiome data. However, given that certain data types occasionally detracted from algorithm accuracy, we found that when combining all factors (genes, CAGs, taxa, pathways) into one model, the resulting classifier was not accurate as they were with individual data types.

We additionally analyzed the annotations of the top 100 genes that were most predictive of host characteristics (S4 Table). Most of these genes (72.2% overall) were "hypothetical," or not been assigned an annotation during gene prediction and *ab initio* annotation. Even when using our BioCyc annotations to identify complexes, pathways, transporters, and enzymes by homology, only 117 of these 500 genes (23.4%) could be annotated (S5 Table). The exception to this rule was for the Antibiotic Usage phenotype, which primarily had annotated genes, some of which were associated with common pathways of antibiotic resistance or potentially horizontal gene transfer, like outer membrane assembly (BamA), DNA recombination (XerD), and efflux (YycB). Additionally, while we found many sex-associated features, including two different highly predictive genes with the same annotation, putative copper-transporting ATPase PacS, we were unable to find a consistent biological theme to the associated functions as a whole.

Finally, we sought to further investigate the identified microbiome associations with mode of infant delivery, as our results indicated a substantial increase in prediction accuracy when including microbiome data instead of demographic features alone. While the majority (85/100) of delivery-type associated features had no BioCyc or Prokka-derived annotation, we found a number (3 genes) that mapped specifically to methylation-associated processes. We additionally found two different associated genes to map to the pathway of heparin

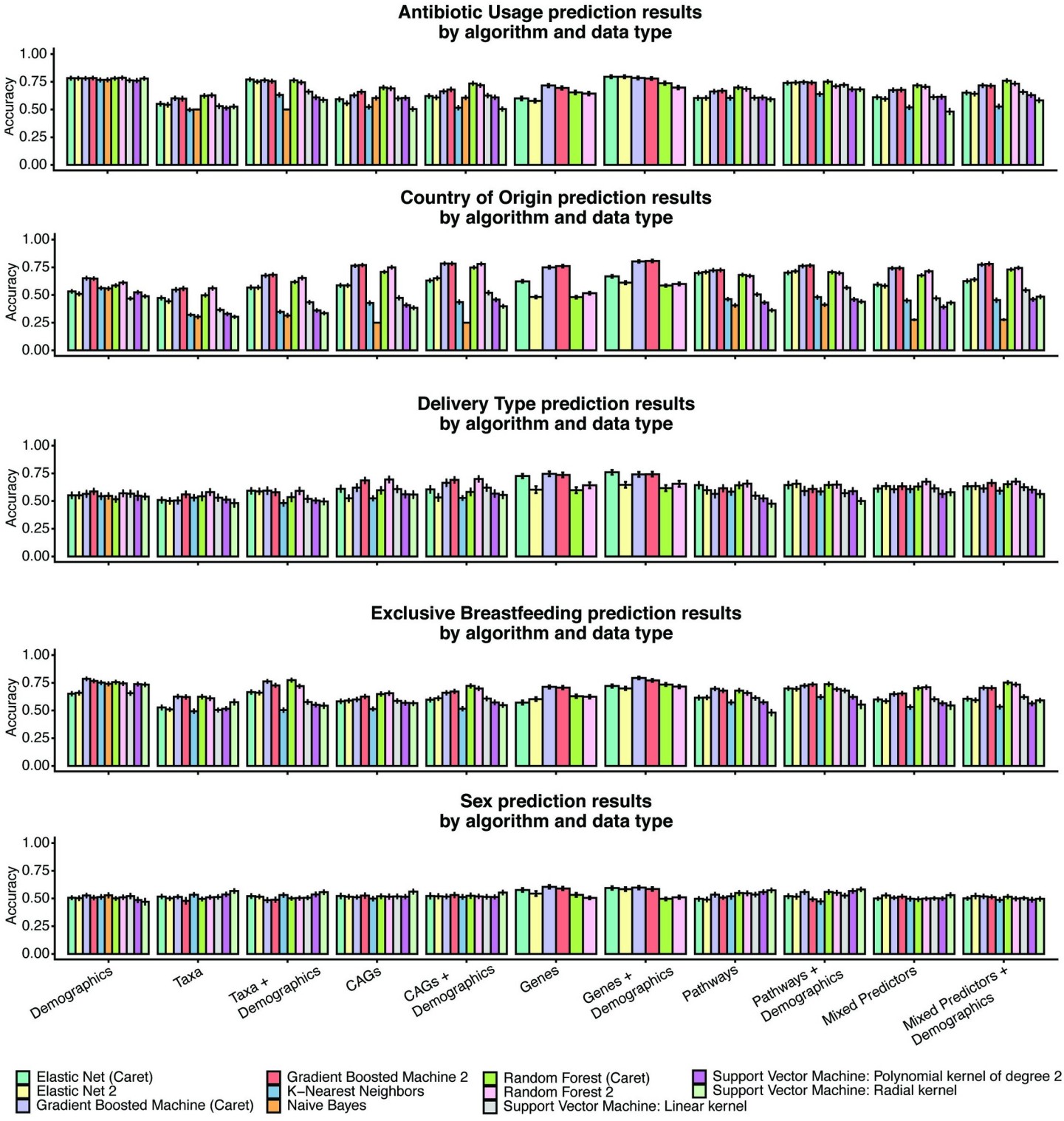

**Fig 3. Classification performances by data type and algorithm for all variables other than age.**

degradation. Overall, we were unable to identify specific biological trends in the association between these genes and delivery type.

## CAGs as predictors of infant age

CAGs and demographics combined were most predictive of age, and Caret's implementation of random forest yielded the best improvement in $R^2$ relative to its demographic baseline ($R^2$ experimental = 0.625+-0.021, $R^2$ baseline = 0.113+0-.014, Root-Mean-Squared-Error = 171.0 +-5.7 days). The combined model (all demographics and phenotypes), as well as the CAG/gene/pathway models also performed similarly, achieving $R^2$ values in the range of ~0.5–0.6 (Fig 4A). On average, non-linear algorithms outperformed linear algorithms for age prediction. Models built with taxonomies and taxonomies/demographics alone fared the worst out of all of the methods. Elastic Nets, K-nearest neighbors, and support vector machines also produced poor results, in some cases with $R^2$ values lower than the baseline model.

The top 25 most relatively important CAGs had Spearman correlations with age ranging from -0.55 to 0.64 (Fig 4B). Each of these CAGs contained 2 to 1,811 genes (median = 9). We hypothesized that identified large portions of entire genomes as well as clusters of accessory genes. To visualize and quantify these associations, we computed the correlations between these CAGs. These ranged from 0.4 to 0.9. We hypothesize that, as has been shown in prior work, this level of co-occurrence demonstrates that some of these CAGs originate from the same biological system, like single species or pathways (S4 Fig) [28].

We annotated the CAGs by assigning their component genes to taxonomies via alignment (S3 Table, S5 Fig). Given the difficulty of assigning individual genes to specific species, the majority of our annotations were at the genus or family level. One set of notable exceptions were 3 highly correlated (Spearman rho > 0.7) CAGs: CAG0751, CAG7244, and CAG1693. They each consisted, in large part, of *Faecalibacterium prausnitzii* genes. The largest was CAG0751 (Fig 4C), which contained 730 genes, 40.9% of which were annotated as belonging to *Faecalibacterium prausnitzii* (another 34.6% were annotated just as *Faecalibacterium*).

CAG0751 was positively associated with age (Spearman = 0.64, Fig 4D, S6 Fig). Only 3 of our top 25 CAGs were negatively correlated. CAG1474 (Fig 4E) was the most extreme (Spearman = -0.55). It had limited annotations, though the 95 genes (56%) that could be taxonomically classified mapped to the family *Enterobacteriaceae*. The other two CAGs negatively associated with age, CAG1188 and CAG1104, had respective Spearman correlations with age of -0.24 and -0.28. They mapped predominantly to *Bifidobacterium* (S5 Fig), and were in turn highly correlated with each other (Spearman = 0.89). Finally, we additionally confirmed that the directionality of these correlations was the same across all four cohorts, implying that our age predictors were robust to variation in geography and other demographic characteristics (S7 Fig).

## Models using MetaPhlAn2 and CAGs identified biologically similar predictors of age

While CAGs outperformed MetaPhlan2 in terms of prediction performance (with maximal $R^2$ values of 0.312 and 0.625, respectively), we found that for their best-performing models, their most age-associated features had overlap in taxonomic classification (S6 Table). For example, *Faecalibacterium prausnitzii* was the most predictive MetaPhlAn2 feature, followed by an unclassified *Oscillibacter* species. To further measure concordance between highly important CAG and MetaPhlAn2 predictors, we built a correlation matrix between our top 25 most predictive CAGs and MetaPhlAn2 annotations. The *Faecalibacterium prausnitzii* MetaPhlAn2 annotations were moderately to strongly correlated (Pearson > 0.4) to different clusters of

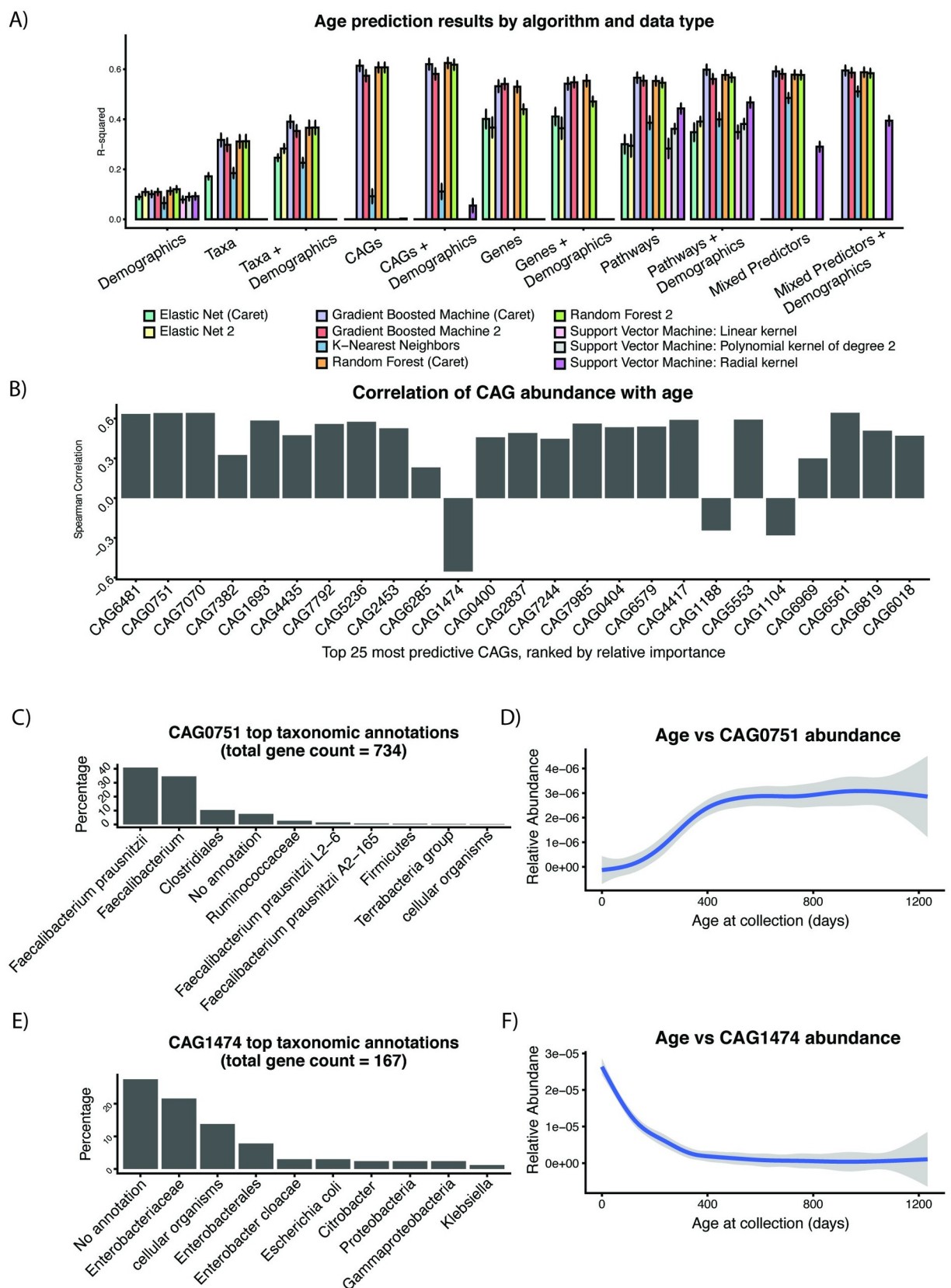

**Fig 4.** A) Performance of metagenomic predictors of infant age by algorithm and data type. B) Correlation of top 25 most predictive CAGs with age C)-F) Taxonomic breakdown and spline fits (relative to age) for a representative positively-associated and negatively-associated CAG.

CAGs (e.g. CAG0761, CAG7070, CAG7244) with phylogenetically identical annotations (S8 Fig, S6 Table).

Other notable consistent annotations for species and CAGs included *Lachnospiraceae* species and *Bifidobacterium ruminantium*. The other highly important large CAGs (>500 genes), annotated predominantly as *Oscillospiracea* (e.g. CAG0404) and *Ruminococcus*, with the former being also highly correlated to a similar MetaPhlAn2 annotation (S8 Fig, S6 Table). We claim these results overall demonstrate that while reliance on reference genomes prevents MetaPhlAn2 output from building accurate predictors, it was still able to pick up on key biological signals from the data.

## Discussion

Here, we demonstrate how variation in modeling approach drastically changes prediction accuracy for a range of human phenotypes and the microbiome. This provides a cautionary note in developing new microbiome-host interaction studies, analyzing existing microbiome studies, and finally, accurately interpreting findings from these studies. We show that there is "no free lunch" in algorithmic choice for microbiome prediction tasks, and that a range of algorithms should be tested to gauge robustness. Second, we demonstrate that while it is challenging to find an optimal model for all phenotypes and data types or even identify why some models outperform others, gene-level, reference-free data outperform reference-based approaches in prediction of human host phenotype. Reference-based pathway-level features were never, in our analysis, the most effective at predicting phenotype when compared to the other data types. Third, we leverage our comprehensive pipeline to identify phenotype-associated microbiome features for infant age, sex, breastfeeding, geography, delivery type, and past antibiotic usage. Fourth, based on the similarity in biological interpretation between our MetaPhlAn2-based and CAG-based predictors, we show that reference-based approaches can certainly at times yield biologically meaningful associations when the ultimate goal is hypothesis generation and not prediction accuracy.

As readers will know, there is currently a proliferation in the types of machine learning algorithms on the analytic market, and choosing or interpreting the "optimal" one is not clear. Further still, the multiple models to choose from can result in phenomena akin to "Vibration of Effects", whereby model choice influences conclusions and inference (and is not made clear to the reader) [11]. Further still, "researcher degrees of freedom"—the analytical choices made by analysts—are known to influence conclusions. For example, in a "many analysts, single dataset" scenario, Silberzahn, Nosek, and colleagues found a variety of models/methods used by different analytic teams influenced conclusions [29]. This is especially a challenge in metagenomics, whereby costly followup biomedical investigations depend on model output. In our investigation, we identified that of all the algorithms tested, GBMs were the most consistently effective at classification/prediction. In some cases, though, no clear winner emerged, and the best choice varied on a host phenotype-by-phenotype basis. For some phenotypes such as age, the information contained in the microbiome seems to be non-linear. The use of non-linear models such as the GBM may greatly improve performance over linear models such as the elastic net. For other phenotypes, the information encoded in the microbiome seems to be linear, as no significant difference was observed between the ENs and the tree based methods. In this situation, the linear methods might be preferred because of their interpretability. At times, even different implementations of the same algorithms (i.e. Random Forests and GBMs)

yielded differing predictive results. We observed some with negative prediction performances (S2 Text). We hypothesize that these poor performances arose in some cases for outliers present in the training folds, and that algorithms robust to the presence of such outliers (GBMs, RF) were not strongly affected, unlike ENs and SVMs. Finally, negative performances could also be observed on every testing fold for some models (e.g SVM with polynomial kernel), which suggests that the model simply fails to generalize.

We reported biological associations for our most robust phenotype-associated findings, including age, delivery type, sex, and antibiotic usage. We note that biological interpretation is going to be dependent on microbiome data type in addition to model choice. Specifically regarding age, we claim that we have identified portions of a patterned, predictable microbial succession early in life [30]. Notably, this has been demonstrated in the past to be the case for one of our top hits for age association, *Faecalibacterium prausnitzii* [31]. That being said, we hesitate to use the functional annotations ascribed to our genes, pathways, and CAGs of interest to form biological hypotheses, as observational, *in silico* analysis of lists of functional characterizations will lead to spurious, or at the very least, nebulous conclusions. For example, pathway annotations can be fraught with the challenge of spurious assignment to function due to sequence homology [32]. In future work, alternative pathway annotation methods that are not reliant on homology alone (e.g. pFam, eggNOG) should be tested in a similar approach to ours [33,34]. Overall, we hope that the individual associations we have identified—and not necessarily any list of functional characterization—can be followed in biological assays in future work. On this note, we are additionally interested in how historical antibiotic usage is identifiable in patient cohorts. This "antibiotic scarring" could potentially provide valuable insight into the long-term effects of medication usage on the microbiome. Finally, future expansions on our analysis could include investigating how single nucleotide variation in genes–or alternative taxonomic binning approaches–function as predictors of host phenotype [35,36].

We hypothesize that CAGs' success in predicting age is likely due to *de novo* assembly capturing genes that are not present in reference databases, meaning that, co-occurrence clusters of those genes capture representations of the "true"gut microbe core genomes, accessory genomes, or functionally linked genes (as many of our CAGs are not the size of complete microbial genomes). We hypothesize that due to the shifting of bacterial genes through rapid evolution and horizontal gene transfer, [18] a genome in a database is unlikely to be representative of what is found in nature. Recent work has demonstrated this to be the case and, moreover, shown the importance of strain-level variation in affecting host phenotype. [2,12] That being said, though, a potential downside of *de novo* assembled data types is that gene-level variation may be so subject-specific that predictors of host phenotype in one cohort may not reproduce well in other cohorts. Additionally, *de novo* assembly results in substantial numbers of "hypothetical," unannotated genes. In our study, up to 72% were hypothetical, but were predictive of host phenotype. While not exactly a "black box", these require functional validation in order to assess biological plausibility.

Analytically, associations with age, sex, and other demographic features indicate the potential confounding of these variables when associating individual microbiome features with host phenotype. We recommend, for investigations uncovering human disease-associated microbial features, that study designs incorporate age and sex as stratifying or adjustment variables in their analysis or use of these variables in a "baseline" model to compare the prediction accuracy.

In summary, it is of fundamental importance—in the era of commoditized computing—to determine if variation in classification due to model choice is a biological finding, a function of the training/testing process, or specific to the data at hand. Therefore, we recommend, before uptake of even more complex analytic techniques to interpret metagenomes, that simpler

techniques for these complex data, such as the elastic net, should be implemented alongside any more complicated one (ie, GBMs or RFs) to ensure biological comprehensibility. With increased algorithmic complexity, the interpretability and reproducibility of results can suffer if the approach implementation, notably the hyperparameters, are not well understood nor searched. It additionally is essential to select microbial features (ie, gene vs. CAG vs. taxonomy vs. pathways) that are compatible or enable the end-goal of the experiment. For example, while genes may build better predictors than species-level data, the latter may be more appropriate if prediction accuracy is not a priority. These considerations relating to feature and model choice will allow for robust results essential for the ultimate goal in host-microbiome association studies such as the development of clinically relevant findings.

## Methods

### Data collection

We downloaded sequencing data (with adapter contamination removed) from the Diabimmune cohort website (https://pubs.broadinstitute.org/diabimmune) and EBI accession number ERP005989. In total, we aggregated 1,570 samples.

### Assembly and construction of non-redundant gene catalogue

We assembled reads (which had been filtered to remove human and adapter contamination) into contiguous sequences (or, "contigs") with the MEGAHIT V1.1.2 (—presets meta) assembly software. We identified genes on our contigs using Prokka V1.12 (—cpus 0—addgenes—metagenome), and then ran CD-HIT-EST V4.6.8 with a 95% identity cutoff (—n 10—c .95 -aS .9 -S .9 -M 0 -T 0) [37–39]. In accordance with the literature, we removed short genes (less than 100bp that had no homolog in NCBI's NR) database from the gene catalog after clustering [28].

### Gene-level abundance quantification and co-abundance clustering

We aligned raw read data to the non-redundant gene catalog using Bowtie2 V2.1.0 on the default settings, and, using a published approach, normalized read counts on a per gene basis by length of gene and total depth of coverage within a sample [40,7]. We clustered co-abundant genes into CAGs using the Canopy clustering algorithm on the default settings (repository link: https://bitbucket.org/HeyHo/mgs-canopy-algorithm/src/master/) [28]. Canopy clustering grouped gene units based on Pearson correlation of 0.9 or greater (parameters:—max_canopy_dist 0.1—max_close_dist 0.4—max_merge_dist 0.1—min_step_dist 0.005—max_num_canopy_walks 5—stop_fraction 1). We computed CAG abundance by taking the 0.75 quantile relative abundance value of all genes within a CAG.

### Species-level annotation and abundance quantification

We ran MetaPhlAn2 V2.1.0 with the default parameters to identify species and their abundances in our data [41].

### Pathway annotation

We aligned our gene catalog to the BioCyc 23.0 Tier 3 database using Diamond V0.9.18 with the default e-value cutoff in addition to a percent identity cutoff of 95% [42]. We allowed twenty alignments per gene (-k equal to 20). Similarly to CAGs, we computed pathway abundance by taking the 0.75 quantile relative abundance value of all genes aligned to a pathway.

The reference databases we used for our pathways alignments can be downloaded from
https://figshare.com/articles/BioCyc_Tier3_Diamond_index_and_mapping_files/9964241.

## Identification of genes for testing

We first identified CAGs that were identified as highly phenotype-associated for our top per-
forming algorithm for that phenotype. We extracted the individual abundances genes from the
top ten most predictive CAGs and then tested the capability of that gene set to predict host
phenotype. As with the other data types, we filtered genes out that did not have bonferroni-
corrected statistically significant univariate associations with phenotypes. We did these filter-
ing steps for two reasons 1) it prevented us from having to test the 10 million genes in our
dataset, which was computationally challenging and 2) it allowed us to determine if the averag-
ing of abundances required to compute CAG abundance reduced predictive signal. We addi-
tionally note that this approach is not without drawbacks: specifically, we are implicitly
subsetting our gene sets to genes that are likely to already be associated with disease.

## Data preprocessing

To obtain a complete matrix, we removed the samples for which one of phenotypes was miss-
ing or where assembly failed (therefore resulting in no CAG or gene level data for that sample).
We were left with 1,219/1,570 (77%) samples for the taxa analysis (excluding a subset samples
from the infants' mothers) and 1,181/1,570 (75%) samples for the CAG analysis (more were
left out in the gene-derived data due to failed assemblies). We converted the category variables
into dummy variables when we used them as predictors, and as factors when we used them as
targets. We split the data into ten folds to perform a "nested" cross validation. Because some
individuals were sampled several times, we ensured that samples from the same individual
were all attributed to the same fold. This way, we prevented the algorithms from taking advan-
tage of the personal signature of each individual to spuriously increase the testing accuracy.
We stratified this sampling procedure for all categorical target variables (sex, country, delivery
type, breastfeeding status, antibiotic usage).

For each fold, we filtered out the taxa or CAGs or pathways that were not significantly asso-
ciated with the target variable. For each taxa/CAG/pathway, we ran a linear regression or a
logistic regression to predict the phenotype of interest from this unique variable. We report
the total associated microbial features in S7 Table. Specifically, after scaling each variable by its
standard deviation, we associated each taxa/CAG with the target phenotype and estimated the
coefficient, standard error, and p-value. Since the goal was dimension reduction, we used a lib-
eral p-value of 0.05 to select taxa/CAG/pathway to input to the next step. If more than 1,000
variables were associated with the phenotype (p-value < 0.05), we only selected the 1,000 vari-
ables whose regression coefficients were the largest to limit the amount of computing
resources necessary to analyze the data. If strictly less than 1,000 variables were significantly
associated with the phenotype, we selected the 1,000 most significantly associated with it
ranked by smallest to largest p-value. Finally, if the number of predictor variables was smaller
than 1,000, we selected all of them (taxa analysis). For the classification tasks, we desired to
correct for class imbalance, so we weighed the samples in each class inversely proportionally to
their proportion in the dataset.

## 10 fold nested cross validation

A visualization of 10 fold nested cross validation is depicted in S3 Fig. We separated the entire
dataset into 10 folds, setting one aside as a final testing fold. We then iterated through each
fold, using each as a sub-testing set once to tune the model. We then tested our final model on

the testing fold we set aside initially. We repeated this process such that every fold was the final testing set once. To avoid bias of Root-Mean-Squared-Error/$R^2$ due to repeated measures, we ensured that the training and testing folds did not have the same individuals in them. To determine our final accuracy measurements, we tested the final model (built from the combination of each test fold) on the entire dataset.

### Hyperparameter tuning

We queried 7,865 models, which were combinations of target phenotypes (n = 6), predictor types (n = 4), algorithms (n = 8), and folds (n = 10) for the cross-validation. We automated the tuning of hyperparameters to consider this space of models.

We used a "pseudo-gradient" descent on the hyperparameters (see steps below). For each algorithm, the hyperparameters we tuned can be found in S8 Table as well as on our web resource. For every Caret algorithm as well as for the Elastic Net 2, we proceeded in the following three steps.

1. Coarse grid initialization
   We explored a coarse grid of hyperparameter values and selected the best hyperparameters combination evaluated on the metrics below. This grid was defined by the stride associated with the hyperparameter. For each hyperparameter, the initial values covered by the grid can be found in S5 Table. The table gives the two extremum values covered, as well as the stride associated with the initial grid. Then we applied the transform function also mentioned in S5 Table to every element of the grid to obtain the actual values to explore for the hyperparameter.

2. Pseudo gradient descent exploration
   If the optimal value for at least one of the hyperparameter values was not a local minimum, we kept exploring the hyperparameter values until the value of the hyperparameter corresponding to the highest performing model was a local minimum. As we explored values outside the initial grid, we only considered the values for the other hyperparameters that were involved in combinations of hyperparameters that gave a performance equal to the best prediction performance. During this exploration of hyperparameters, the model would occasionally fail due to memory errors stemming from specific hyperparameters. In that case, the limit of the explorable values for this hyperparameter was reset accordingly. We looped through every hyperparameter, the value was either at a local minimum of the loss function, or equal to one of the limits set for the explorable range of values for this hyperparameter.

3. Fine tuning:
   Third, we fine-tuned the hyperparameters by exploring a finer grid of values around the best selected values. We set the range of values to explore in two ways. For an hyperparameter, if more than one value was involved in the best performance obtained during the previous search, we explored values between the two extrema of these values. If the best performance was only observed with a specific value of the hyperparameter, we explored values between the best value and the second-best value in one direction, and between the best value and half a stride in the other direction. The number of values to explore in this space was determined by the 'N fine tuning' column in S5 Table and was chosen based on the computing resources available to us.

## Metrics

To tune the regression tasks (Age prediction) we used $R^2$. For binomial classification tasks, we used the area under the curve of the receiver-operator curve (ROC). For multinomial classifications, we used the accuracy. We additionally compared the $R^2$/ROC of the models containing microbiome data to a "baseline" model, which only contained the demographic information not being predicted (age, sex, or country of origin. These "control" models were meant to determine if most of our predictive power came from the demographics variables. If this were the case, we would not see a significant difference between the predictive accuracy obtained on the "demographics only" model and the models that leverage microbiome information.

## Complications

Some of the models were able to compute without error and output a performance indicator (such as R-Squared or ROC). However, they would only output NaN when used to generate predictions. This problem was only encountered is less than 0.01% of the models tuned. To prevent this from happening, every time a better set of hyperparameters was found, we tested if the model was able to generate predictions on the training set. If it did not, we did not replace the previous best combination of hyperparameters with the new, better performing combination.

## During testing

Complication:

It happened that a model would only output NaN on the testing set. We looked more closely at this model and were able to identify a single sample that would make the model generate NaN for every sample in the fold, when this sample was present in the fold. Therefore, when a model outputted NAs on the testing set, we reran the model to predict every single sample individually. We then excluded the samples for which NAs were generated from the analysis. To our knowledge, this only affected one sample for two specific models out of the thousands of models run.

For Naïve Bayes, the models could generate the class predictions, but most often outputted NaN for the probability predictions. Because of that, the metrics built on the probabilities (such as ROC or cross-entropy) often have missing values, whereas the metrics computed based on the class predictions (such as accuracy or mean class accuracy) were successfully calculated.

For Gradient Boosted Machine 2:

We set the interaction depth to one, the minimal number of observations per node to ten, and the shrinkage to 0.1. We initialized the number of trees to 1000 and tuned the model a first time. As long as the model's best performance was obtained during the last 10% of the iterations (e.g the last 100 trees if the total number of trees was 1000), and if there was at least a 0.1% improvement between the mean performance of the last 10% of the iterations, and the 10% before that, we reran the model with twice more trees. This allowed us to make sure that we used a large enough number of trees to find a decent minimum of the loss function.

For Random Forest 2:

We set the number of trees to 1001 and we selected the step to be 0.9. If the model was a regression, we selected the sample size to be 0.632*N_rows. Otherwise, we sampled from each category the same number of samples, equal to the number of samples in the smallest category. We selected mtry by performing cross-validation.

## Evaluation of classification performance

We concatenated the testing predictions generated on each of the ten folds, we measured the performance of the algorithms on the entire concatenated testing set, and we bootstrapped this measure 1,000 times to obtain confidence intervals. For example, to estimate the confidence interval on the age prediction $R^2$ of a model, we considered the 1,570 predictions for the 1,570 samples in the dataset. We then took 1,570 samples with replacement from these 1,570 predictions for 1,000 iterations. Each iteration, we calculated the $R^2$ on the sampled dataset. Finally, we calculated the standard deviation on the sampled 1,000 $R^2$ values to obtain an "empirical" confidence interval on the actual value of the testing $R^2$.

## Algorithmic concordance analysis

We aimed to compute the concordance between the ranking of phenotype-associated features across 1) algorithms and 2) data types. In other words, we aimed to determine if algorithms were ranking features similarly in terms of their association with a given phenotype (e.g. Do random forests and elastic nets prioritize the same microbial species in their association with age?). Analogously, we compared the if, for a given phenotype, if Metaphlan2, pathway, or CAG associations were more consistent in terms of relative importance across all algorithms. For CAGs, MetaPhlAn2 output, and pathways, we weighted the ranking of the 1000 variables that went into each prediction by their relative importance/impact on mean squared error/$R^2$. We then computed pairwise Spearman correlation for each vector of weights for each algorithm/data type/phenotype.

## Principal component analysis

We executed our Principal Component Analysis using R's prcomp() function, scaling input columns to have unit variance.

## Code availability

The code, written in a combination bash, Python V2.7, and R V3.44 we used to carry out our project is available at https://github.com/chiragjp/metagenome_ml.

## Supporting information

**S1 Text An overview of our machine learning toolset and additional technical notes on outliers in our results.**
(PDF)

**S2 Text. An overview of our machine learning toolset and additional technical notes on outliers in our results.**
(PDF)

**S1 Table. Table of definitions relevant for this manuscript.**
(XLSX)

**S2 Table. Dataset summary by cohort.**
(XLSX)

**S3 Table. Algorithm and data type prediction and classification accuracies.**
(XLSX)

**S4 Table. Top 100 most predictive genes for breastfeeding status, country of origin, antibiotic usage, country of origin, and sex.**
(XLSX)

**S5 Table. Additional functional annotations for the top 100 most predictive genes for breastfeeding status, country of origin, antibiotic usage, country of origin, and sex.**
(XLSX)

**S6 Table. Taxonomic annotations for top 25 CAGs and MetaPhlAn2 results most predictive of age.**
(XLSX)

**S7 Table. Number of features initially correlated to phenotypes of interest prior to building predictive models (see Methods for more details).**
(XLSX)

**S8 Table. Hyperparameters used in algorithm training.**
(XLSX)

**S1 Fig. Exploratory principal component analysis of all demographic variables, colored by dataset.**
(PDF)

**S2 Fig. The number of genes per CAG in our dataset alongside quantiles of the frequency distribution.**
(PDF)

**S3 Fig. Detailed walkthrough and example of our 10x cross-fold validation strategy.**
(PDF)

**S4 Fig. Correlation matrix between top 25 most age-associated CAGs.**
(PDF)

**S5 Fig. Taxonomic breakdown for the remaining 23 out of the top 25 most predictive CAGs for age that were not shown in the main text.**
(PDF)

**S6 Fig. Spline fits for each or our top 25 most predictive CAGs for age, by age and cohort.**
(PNG)

**S7 Fig. Correlation for top 25 CAGs with age, by cohort.**
(PDF)

**S8 Fig. Correlation between top 25 CAGs and MetaPhlAn2 taxonomies associated with age.**
(PDF)

## Author Contributions

**Conceptualization:** Alan Le Goallec, Braden T. Tierney, Aleksandar D. Kostic, Chirag J. Patel.

**Data curation:** Alan Le Goallec, Braden T. Tierney, Jacob M. Luber, Evan M. Cofer.

**Formal analysis:** Alan Le Goallec, Braden T. Tierney, Jacob M. Luber, Evan M. Cofer.

**Funding acquisition:** Aleksandar D. Kostic, Chirag J. Patel.

**Investigation:** Aleksandar D. Kostic, Chirag J. Patel.

**Methodology:** Alan Le Goallec, Braden T. Tierney, Aleksandar D. Kostic, Chirag J. Patel.

**Project administration:** Aleksandar D. Kostic, Chirag J. Patel.

**Resources:** Aleksandar D. Kostic, Chirag J. Patel.

**Supervision:** Aleksandar D. Kostic, Chirag J. Patel.

**Visualization:** Alan Le Goallec, Braden T. Tierney.

**Writing – original draft:** Alan Le Goallec, Braden T. Tierney, Aleksandar D. Kostic, Chirag J. Patel.

**Writing – review & editing:** Alan Le Goallec, Braden T. Tierney, Jacob M. Luber, Evan M. Cofer, Aleksandar D. Kostic, Chirag J. Patel.

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
