## [Decision Letter · Decision Letter 0]

26 Dec 2019

Dear Dr Patel,

Thank you very much for submitting your manuscript 'A systematic machine learning and data type comparison yields robust metagenomic predictors of infant age, sex, breastfeeding, antibiotic usage, country of origin, and delivery type' for review by PLOS Computational Biology. Your manuscript has been fully evaluated by the PLOS Computational Biology editorial team and in this case also by independent peer reviewers. The reviewers appreciated the attention to an important problem, but raised some substantial concerns about the manuscript as it currently stands. While your manuscript cannot be accepted in its present form, we are willing to consider a revised version in which the issues raised by the reviewers have been adequately addressed. We cannot, of course, promise publication at that time.

Sincerely,

Nicola Segata

Associate Editor

PLOS Computational Biology

Jason Papin

Editor-in-Chief

PLOS Computational Biology

[LINK]

While the reviewers found the paper of potential interest, they all rised substantial concerns. If the authors are able to address all the issued raised by the reviewer, I would be willing to consider a revised manuscript.

Reviewer's Responses to Questions

**Comments to the Authors:**

Reviewer #1: The authors present a systematic comparison of machine learning tools applied to microbiome data.

Overall, this may be interesting, but I'm lacking three major parts:

1. The motivation is not sufficient to explain what we're gaining in this paper.

2. Very much related, the outcome of this paper is unclear to me. It reads more like a grocery list of models and their results, and which model will find which variable, without any profound conclusion that the reader can take home.

3. Personally, I think papers should be accessible to undergraduate students. This paper is not deductive enough. It is difficult to read. It uses terms (and acronyms) without explaining them, and it doesn't tell a story that the reader can follow.

The introduction is lacking some papers that have worked on these problems before, with addressing their successes and limitations. A big portion of the introduction is really results from this paper. As the reader reaches the real results section the motivation is still unclear.

Maybe if such an intro was present, the authors would not have claimed to be the "first to successfully predict infant age and sex from microbiome data". For example, age was predicted in a paper that examined malnutrition from the Gordon lab (Subramanian et al, Nature 2014), and used to find that malnutritioned children had a "younger microbiome age". If the authors meant to say that others may have predicted age, but they are the first to predict both age and sex, then this current text in misleading. Also, again, the motivation of why such a prediction would be useful is greatly missing.

Overall, I think that comparing many tools could be interesting, but current paper does not teach its readers about why some models are better than others, and does not reveal any novel biological insight from this comparison.

Reviewer #2: Review is uploaded as an attachment.

Reviewer #3: Attached

**Have all data underlying the figures and results presented in the manuscript been provided?**

Reviewer #1: Yes

Reviewer #2: Yes

Reviewer #3: No: The supplemental figures appear to be missing.

PLOS authors have the option to publish the peer review history of their article (what does this mean?). If published, this will include your full peer review and any attached files.

Reviewer #1: No

Reviewer #2: No

Reviewer #3: No

---

## [Decision Letter · Decision Letter 1]

13 Apr 2020

Dear Dr. Patel,

Thank you very much for submitting your manuscript "A systematic machine learning and data type comparison yields metagenomic predictors of infant age, sex, breastfeeding, antibiotic usage, country of origin, and delivery type" for consideration at PLOS Computational Biology. As with all papers reviewed by the journal, your manuscript was reviewed by members of the editorial board and by several independent reviewers. The reviewers appreciated the attention to an important topic. Based on the reviews, we are likely to accept this manuscript for publication, providing that you modify the manuscript according to the review recommendations.

Thanks for your revised manuscript. The reviewers faviourably evaluated it. They still have some comments and suggestions to clarify some points and further improve readbility that should be possible to address with a minor revision.

Sincerely,

Nicola Segata

Associate Editor

PLOS Computational Biology

Jason Papin

Editor-in-Chief

PLOS Computational Biology

[LINK]

Thanks for your revised manuscript. The reviewers faviourably evaluated it. They still have some comments and suggestions to clarify some points and further improve readbility that should be possible to address with a minor revision.

Reviewer's Responses to Questions

**Comments to the Authors:**

Reviewer #1: The authors have addressed my previous comments.

I would still consider narrowing down the number of models tested (while moving some to the supp material) so the paper can be more concise and (yet again) more educating to the reader in terms of types of models and their expected behavior.

Reviewer #2: Uploaded as an attachment

Reviewer #3: The authors have addressed my concerns.

**Have all data underlying the figures and results presented in the manuscript been provided?**

Reviewer #1: Yes

Reviewer #2: Yes

Reviewer #3: Yes

PLOS authors have the option to publish the peer review history of their article (what does this mean?). If published, this will include your full peer review and any attached files.

Reviewer #1: No

Reviewer #2: No

Reviewer #3: No
---

## [Editor Report · Decision Letter 2]

21 Apr 2020

Dear Dr. Patel,

We are pleased to inform you that your manuscript 'A systematic machine learning and data type comparison yields metagenomic predictors of infant age, sex, breastfeeding, antibiotic usage, country of origin, and delivery type' has been provisionally accepted for publication in PLOS Computational Biology.

Best regards,

Nicola Segata

Associate Editor

PLOS Computational Biology

Jason Papin

Editor-in-Chief

PLOS Computational Biology

---

## [Editor Report · Acceptance letter]

4 May 2020

PCOMPBIOL-D-19-01912R2 

A systematic machine learning and data type comparison yields metagenomic predictors of infant age, sex, breastfeeding, antibiotic usage, country of origin, and delivery type

Dear Dr Patel,

I am pleased to inform you that your manuscript has been formally accepted for publication in PLOS Computational Biology. Your manuscript is now with our production department and you will be notified of the publication date in due course.

With kind regards,

Bailey Hanna
